

# The role of Central American barriers in shaping the evolutionary history of the northernmost glassfrog, *Hyalinobatrachium fleischmanni* (Anura: Centrolenidae)

Angela M. Mendoza[1,2], Wilmar Bolívar-García[3], Ella Vázquez-Domínguez[4], Roberto Ibáñez[5,6,7] and Gabriela Parra Olea[1]

[1] Departamento de Zoología, Instituto de Biología, Universidad Nacional Autónoma de México, Mexico city, México

[2] Posgrado en Ciencias Biológicas, Universidad Nacional Autónoma de México, Mexico City, Mexico

[3] Departamento de Biología, Grupo de Investigación en Ecología Animal, Universidad del Valle, Cali, Colombia

[4] Departamento de Ecología de la Biodiversidad, Instituto de Ecología, Universidad Nacional Autónoma de México, Mexico City, México

[5] Departamento de Zoología, Smithsonian Tropical Research Institute, Balboa, Panamá

[6] Universidad de Panamá, Panamá

[7] Sistema Nacional de Investigación, Panamá

Corresponding author
Gabriela Parra Olea,
gparra@ib.unam.mx

## ABSTRACT

The complex geological history of Central America has been useful for understanding the processes influencing the distribution and diversity of multiple groups of organisms. Anurans are an excellent choice for such studies because they typically exhibit site fidelity and reduced movement. The objective of this work was to identify the impact of recognized geographic barriers on the genetic structure, phylogeographic patterns and divergence times of a wide-ranging amphibian species, *Hyalinobatrachium fleischmanni*. We amplified three mitochondrial regions, two coding (COI and ND1) and one ribosomal (16S), in samples collected from the coasts of Veracruz and Guerrero in Mexico to the humid forests of Chocó in Ecuador. We examined the biogeographic history of the species through spatial clustering analyses (Geneland and sPCA), Bayesian and maximum likelihood reconstructions, and spatiotemporal diffusion analysis. Our data suggest a Central American origin of *H. fleischmanni* and two posterior independent dispersals towards North and South American regions. The first clade comprises individuals from Colombia, Ecuador, Panama and the sister species *Hyalinobatrachium tatayoi*; this clade shows little structure, despite the presence of the Andes mountain range and the long distances between sampling sites. The second clade consists of individuals from Costa Rica, Nicaragua, and eastern Honduras with no apparent structure. The third clade includes individuals from western Honduras, Guatemala, and Mexico and displays deep population structure. Herein, we synthesize the impact of known geographic areas that act as barriers to glassfrog dispersal and demonstrated their effect of differentiating *H. fleischmanni* into three markedly isolated clades. The observed genetic structure is associated with an initial dispersal event from Central America followed by vicariance that likely occurred during the Pliocene. The southern samples are characterized by a very recent population expansion, likely related to sea-level and climatic oscillations during the Pleistocene, whereas the structure

of the northern clade has probably been driven by dispersal through the Isthmus of Tehuantepec and isolation by the Motagua–Polochic–Jocotán fault system and the Mexican highlands.

## INTRODUCTION

Historical biogeography focuses on the role of the geographic space as a driver of biological processes such as speciation, extinction, and diversification (*Cox, Ladle & Moore, 2016*). Areas with a complex geological history are characterized by the appearance and disappearance of multiple barriers and corridors in their history. These barriers may significantly affect the gene flow of resident species, leading to allopatric speciation by vicariance, whereas corridors may lead to species dispersal and colonization of new areas (*Noss, 1991*). The use of molecular data for the reconstruction of species relationships, the development of new methods for biogeographic analyses, and the increase in geological studies in complex regions have revolutionized the understanding of such biological processes (*Ronquist & Sanmartín, 2011*). Biogeographic studies have integrated information regarding the relationships within or between closely related taxa, providing valuable opportunities to understand how patterns of biodiversity may have been shaped, even at short time scales (*Crawford, Bermingham & Polanía-S, 2007*; *Streicher et al., 2014*).

Central America is a region with a rather complex biogeographic history and high diversity of habitats and species (*Myers et al., 2000*; *Cavers, Navarro & Lowe, 2003*; *Iturralde-Vinent, 2006*; *Daza, Castoe & Parkinson, 2010*). The region is delimited to the north by the Isthmus of Tehuantepec (IT) in Mexico and to the south by the Andes in Colombia (*Gutiérrez-García & Vázquez-Domínguez, 2013*). The geological landscape of Central America has been continuously modified, especially during the last 15 million years (Ma), by major events including the emergence of the Panama Arc (13–15 Ma, *Montes et al., 2015*), the posterior closure of the Panama Isthmian land bridge when it ceased to function as a seaway (∼9–10 Ma, *Montes et al., 2012a*; *Montes et al., 2012b*; *Ramírez et al., 2016*), and the posterior global climatic transitions during the Plio-Pleistocene (*Montes et al., 2015*). These events triggered the Great American Biotic Interchange, or GABI, involving the replacement of native taxa (extinctions) and the establishment and diversification of colonizing taxa (speciation) on both continents (*Marshall et al., 1982*; *Stehli & Webb, 1985*). Ample phylogeographic research in this region has allowed the effects of geomorphology, topographic barriers, volcanic activity, large climate changes, intermittent connections, and corridors on the biota to be described, aiding in our understanding of the influence of past events on the patterns of genetic structure and the geographic distribution of birds (*García-Moreno et al., 2004*; *Cadena, Klicka & Ricklefs, 2007*; *Arbeláez-Cortés, Nyári & Navarro-Sigüenza, 2010*), plants (*Cavers, Navarro & Lowe, 2003*; *Ornelas, Ruiz-Sánchez & Sosa, 2010*; *Cavender-Bares et al., 2011*), reptiles (*Hasbún*

*et al., 2005*; *Venegas-Anaya et al., 2008*), mammals (*Eizirik et al., 2001*; *Ordóñez Garza et al., 2010*; *Pérez-Consuegra & Vázquez-Domínguez, 2017*), and amphibians (*Mulcahy, Morrill & Mendelson, 2006*; *Crawford, Bermingham & Polanía-S, 2007*; *Wang, Crawford & Bermingham, 2008*; *Hauswaldt et al., 2011*). As a result, diverse geological factors and major barriers have been more frequently correlated with the evolutionary history and dispersal of species (*Bagley & Johnson, 2014*).

Amphibians are excellent systems for studies in which geological and environmental histories are inferred at fine scales, due to their ecology, particularly regarding their terrestrial habits, intolerance to salt water (*Beebee, 2005*), and marked niche conservatism (*Smith, Stephens & Wiens, 2005*; *Wiens et al., 2006*), as well as the restriction of the particular habitats of many species (e.g., *Savage, 2002*). However, evaluation of the impact of barriers on the phylogeographic patterns of this taxon, extending across the entire Central American region, has been precluded because most amphibians have small ranges.

Glassfrogs (Centrolenidae) comprise a diverse family endemic to the Neotropics, with numerous species and high levels of endemism, mainly distributed among the Northern Andes and Central America regions (*Guayasamin et al., 2009*; *Castroviejo-Fisher et al., 2014*; *Mendoza & Arita, 2014*). Studies on the dispersal capability of glassfrogs are limited, but these frogs are known to be restricted to streamside habitats (*Ruiz-Carranza & Lynch, 1991*) and to show site fidelity (*Valencia-Aguilar, Castro-Herrera & Ramírez-Pinilla, 2012*) and low mobility, with potential genetic subdivision and restricted gene flow (*Delia, Bravo-Valencia & McDiarmid, 2017*; *Robertson, Lips & Heist, 2008*). The glassfrog *Hyalinobatrachium fleischmanni* (Boettger, 1893) has one of the widest distributions, ranging from Guerrero and Veracruz states in Mexico through the lowlands of Central America, to the southernmost limit of its distribution in Ecuador (*Cruz et al., 2017*). Males of the species call from vegetation along the margins of streams, and egg masses are usually laid on the underside of leaves over a stream. This species exhibits site fidelity and parental care by males, who attend one or more clutches at the same time (*Delia et al., 2010*; *Savage, 2002*; *Barrera-Rodríguez, 2000*). Tadpoles fall from vegetation into the water, where they develop; they are apparently fossorial, living buried in the leaf litter and bank substrate of streams (*Villa & Valerio, 1982*). Related species (i.e., *Hyalinobatrachium tatayoi*, *H. carlesvilai*, *H. mondolfii*, *H. kawense*, and *H. munozorum*) are distributed in different regions of South America, including the northern and central Andes, Guiana shield, and Amazon basin, where previous analyses have suggested an Andean origin for *H. fleischmanni* (*Castroviejo-Fisher et al., 2014*).

Considering its wide distribution, coupled with its site fidelity, *H. fleischmanni* is an ideal organism for studying the role that Central American geographic barriers have played in the dispersal patterns of lowland glassfrog species. In the present study, we had the following objectives: (1) to reconstruct the historical biogeography that has shaped the evolutionary history of *H. fleischmanni*, including dispersal or vicariance events and time of divergence; (2) to evaluate the possible presence of multiple isolated lineages within *H. fleischmanni*; and (3) to identify the impact of recognized geographic barriers on the genetic structure and phylogeographic patterns of *H. fleischmanni* over time. Based on known information about the species, we tested the hypothesis that *H. fleischmanni* had
a South American origin and subsequently dispersed into the Central American lowlands after the closure of the Isthmus of Panama. Additionally, we hypothesized that the dispersal of this species in Central America has been limited by various high mountain ranges acting as barriers and that changes in sea level during the Pleistocene had an impact on the genetic structure of the lowland populations (*Bagley & Johnson, 2014*). Hence, our prediction is that the current genetic structure of *H. fleischmanni* reflects patterns of vicariance events driven by dispersal barriers.

## MATERIAL AND METHODS

### Tissue sampling

Genetic material was obtained across the entire distribution area of the species, from both museum collections and fieldwork (Fig. 1). Fieldwork was performed during the rainy season, in which at least three individuals were collected at each locality. Specimen collection permits were provided by the Ministerio de Medio Ambiente, Colombia (Resolution 120 of 24 August 2015) and the Secretaría del Medio Ambiente y Recursos Naturales, Mexico (office number 00947/16). Captured specimens were euthanized with a 20% lidocaine hydrochloride (Xylocaine) injection, and all efforts were made to minimize suffering. Liver or muscle tissues were collected in the field and were stored in an RNAlater solution until their use in the laboratory. Specimens were fixed with 10% formalin, stored in 70% ethanol and deposited in biodiversity collections at public research institutions in each country.

### Molecular techniques

DNA was obtained from muscle and liver tissues following the phenol-chloroform extraction protocol (*Sambrook & Russell, 2006*). The quantity and quality of the DNA were verified in 1% agarose gels and by measuring absorbance using a NanoDrop spectrometer (Thermo Fisher Scientific Inc., Wilmington, DE, USA). Amplification of the mitochondrial COI (658 bp), 16S (895 bp), and ND1 (961 bp) genes was performed following the protocols described by *Guayasamin et al. (2008)*. PCR products were visualized with agarose gels and purified according to the EXO-SAP protocol (GE Healthcare, Chicago, IL, USA). DNA sequences were obtained with the BigDye Terminator Cycle Sequencing kit (Applied Biosystems). Sequences from *H. tatayoi* from Venezuela (*Guayasamin et al., 2008*; *Castroviejo-Fisher, Ayarzagüena & Vila, 2007*) were also included. The sequences were assembled, manually edited and aligned with Geneious 9.1.2., and the three genes were concatenated by using the *cbind* function in the *ape* package in R (*Paradis, Claude & Strimmer, 2004*).

We are aware that analyses based on mitochondrial DNA (mtDNA) provide a limited view of species evolution (i.e., matrilineal inheritance), but the available genetic material from nuclear Central America is very scarce since there has been limited or no sampling in the region (due to a lack of funding as well as logistical and even bureaucratic difficulties) compared to that in northern or southern countries. Therefore, we chose to compile a database including gene sequences available from GenBank and BOLD databases with our own sequences to perform our analyses. In this way, our study encompasses the entire
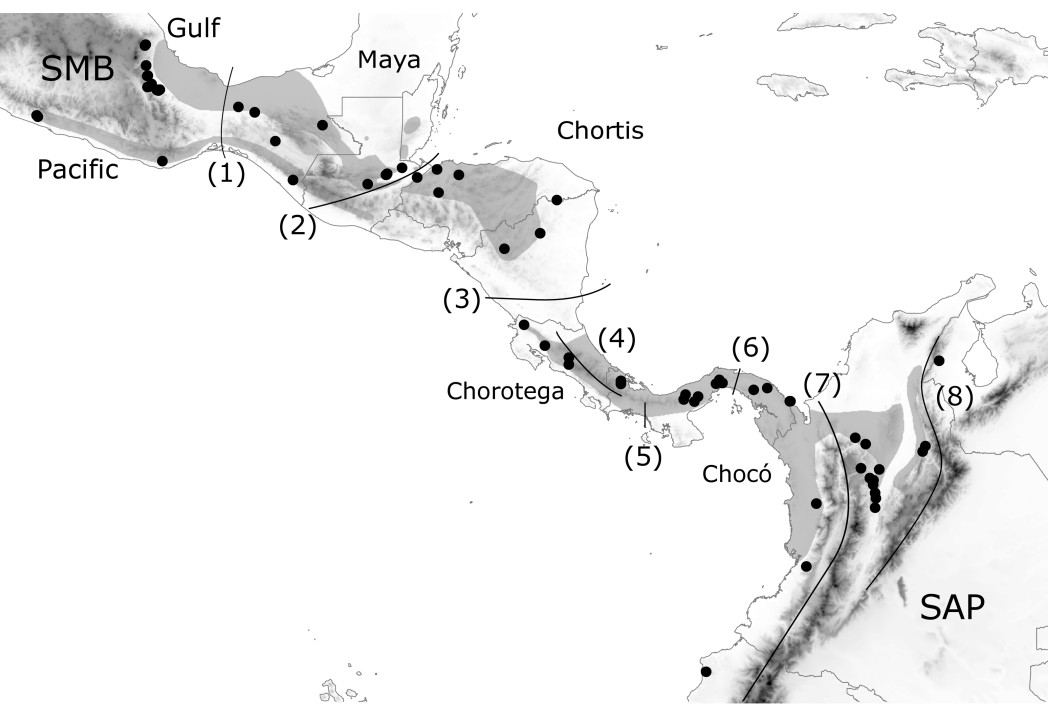

**Figure 1 Geographic distribution of *H. fleischmanni* samples.** Geographic distribution of *H. fleischmanni* samples. Main geological blocks, delimited by geological barriers, are shown. Sample origins are indicated by black dots, while gray polygons show the species distribution according to IUCN. SMB, South Mexican block, SAP, South American Plate. The evaluated barriers are numbered as follows: 1, Isthmus of Tehuantepec, 2, Motagua-Polochic-Jocotán fault system, 3, Hess Escarpment, 4, Talamanca Range, 5, Western Panama Isthmus, 6, Central Panama Isthmus, 7, Northern Andes and 8, Eastern Cordillera.

geographic distribution of *H. fleischmanni*, allowing us to evaluate the events shaping the evolutionary history of the species.

Additionally, mtDNA is a robust indicator of population structure and historical biogeographic barriers and can be used to detect cryptic diversity (*Gehara et al., 2014*; *Dudoit et al., 2018*; *Lait & Hebert, 2018*). Mitochondrial DNA markers have been used in a variety of Central American studies to decipher genetic structuring (for a review, see *Bagley & Johnson, 2014*), allowing us to make direct comparisons with previous studies from this region.

## Data analysis
### Identification of landscape barriers to dispersal
We defined *a priori* a set of possible geographic elements based on the four geological blocks across Central America (Maya, Chortis, Chorotega, and Chocó) (*Marshall, 2007*; *Gutiérrez-García & Vázquez-Domínguez, 2013*) as well as their northern and southern limits: the southern Mexico Block (SMB) west of the IT and the South American Plate (SAP) east of the Andes, respectively. The effects on species dispersal were evaluated for three highland barriers: the Motagua–Polochic–Jocotán (MPJ) fault system, the Talamanca Cordillera, and the Andes range, which separates the Northern Andes (Western and Central

Cordilleras) and the Eastern Cordillera (including Serranía del Perijá). Three previously recognized barriers for lowland amphibians (the Hess Escarpment, HE; western Panama Isthmus, WPI; and central Panama Isthmus, CPI) were also tested as possible factors during Pleistocene sea-level oscillations (Fig. 1). Each barrier was tested by four spatial and non-spatial analyses, and supported barriers were defined as those detected by at least three of the analyses.

The first step was to perform spatial and clustering analyses that are commonly applied to mtDNA sequences. The spatial locations of genetic discontinuities were estimated with Geneland (*Guillot, Mortier & Estoup, 2005*), which estimates the number of populations within the geographical area of interest, maps borders between populations, assigns individuals to populations, and detects possible migrants. We ran the model in R under the correlated allele frequency model, without uncertainty regarding spatial locations. We generated $10^5$ iterations to a thinning of 100, with the maximum rate of the Poisson process fixed as the number of individuals, and the generated map borders were compared with our hypothesized barriers. Additionally, we performed a spatial principal component analysis (sPCA; *Jombart et al., 2008a*) using the *adegenet* package in R (*Jombart, 2008b*) for which we constructed a neighbor-distance net among all coordinates and tested for significant, geographically correlated genetic structures along the main axis based on a global randomization test. We extracted the values of the first two sPCA components and generated polygons for each interbarrier set of individuals to test if the populations adjacent to each barrier were clustered (non-overlap) or displayed as a continuum (partially or completely overlapping). Previous studies have suggested multivariate ordination analyses as an alternative to Bayesian algorithms because they do not make any assumptions about the underlying population genetic model (*Jombart, Devillard & Balloux, 2010*). Additionally, we performed a non-spatial test by estimating the Nei's pairwise $F_{ST}$ value between adjacent regions and estimated significance by a Monte Carlo test based on 999 permutations with the *hierfstat* package (*Goudet, 2005*); because we did not have data for the three mitochondrial genes for all samples in all localities, these analyses included missing values. Additionally, we used the alignments per gene to calculate diversity indices and to perform additional clustering analyses that do not support missing values in the concatenated sequences (see Supplemental Information 1).

## Phylogenetic analyses

Since phylogeographic breaks can be detected in the form of phylogenetic splits between mostly distinct geographical lineages (*Bagley & Johnson, 2014*), the sequences of all genes were concatenated, and a phylogenetic tree was estimated at the intraspecific level by implementing both likelihood analysis in RAxML (*Stamatakis, 2006*) and a Bayesian inference approach in MrBayes (*Ronquist & Huelsenbeck, 2003*). We rooted our phylogeny using the species *Hyalinobatrachium carlesvilai, H. mondolfii, H. chirripoi* and *H. colymbiphyllum* as outgroups. A list of the specimens and GenBank accession numbers included in this study is presented in Table S1. The best evolutionary model for each noncoding region (16S) and for the coding genes (COI and ND1) was evaluated using PartitionFinder 2 software (*Lanfear et al., 2016*). Maximum likelihood analysis

was conducted using 10,000 rapid bootstrap analyses, the GTR+Γ evolution model and summarized support for the best tree. For Bayesian inference, we ran two independent analyses for 12 million generations, sampling trees and parameter values every 1,000 generations. Burn-in was set to 25%, and the first 3 million generations were therefore discarded.

### Divergence times and Bayesian spatiotemporal diffusion analyses

To estimate diversification times for the different *H. fleischmanni* mitochondrial lineages, we employed Beast 1.6.2 (*Drummond & Rambaut, 2007*). The time to the most recent common ancestor (MRCA) for the main lineages was calculated via Bayesian Markov chain Monte Carlo (MCMC) searches. The ultrametric tree was inferred *de novo* using the same partition substitution models. In the absence of a fossil record for glassfrogs, we based our analysis on previously published divergence times. We used three stem ages for *Hyalinobatrachium* species as calibration constraints, following *Castroviejo-Fisher et al. (2014)*. The most recent calibration point was placed at 2.42 Ma (confidence interval CI [1.63–3.37]), representing the divergence between *H. fleischmanni* (USNM 559092) and *H. tatayoi* (MHNLS 17174). The following calibration node was placed at 7.65 Ma (CI [5.93–9.63]), representing the divergence between *H. fleischmanni-H. tatayoi* and *H. carlesvilai*, and the most ancient calibration point corresponded to the divergence between *H. fleischmanni* and *H. mondolfii* (8.4 Ma, CI [6.68–10.52]). The calibration points on inner nodes in *Castroviejo-Fisher et al. (2014)* were based on a geological vicariance-based strategy, which requires additional precautions (*Kodandaramaiah, 2011*) when compared to the fossil-based calibration approach. For more detail, see the section on divergence time estimates and Appendix S1 in *Castroviejo-Fisher et al. (2014)*. We implemented an uncorrelated lognormal relaxed molecular clock, and trees were sampled every 1,000th iteration for 100,000,000 generations, with 20% of the initial samples being discarded as burn-in, after empirical assessment of appropriate chain convergence and mixing with Tracer 1.7 (*Rambaut et al., 2018*). We constructed the historical demography of the major clades obtained from the phylogenetic results, using Bayesian skyline plots that estimate the posterior distribution of population sizes (*Drummond et al., 2005*).

To reconstruct the ancestral distribution and spatial dispersal of the species, we performed a Bayesian spatiotemporal diffusion analysis in BEAST (v.1.8.4), assuming continuous spatial diffusion with a time-heterogeneous random walk model ("Relaxed Random Walk", RRW, (*Lemey et al., 2010*). For this analysis, we used a subset of 34 samples with data for all three genes plus samples lacking some genes but originating from intermediate localities, encompassing the entire distribution of the species. We applied a normally distributed diffusion rate, a coalescent Bayesian Skyride model, and SRD06 substitution models (*Shapiro, Rambaut & Drummond, 2005*). We used the jitter option under the TraitLikelihood statistic with a parameter value of 0.1. To summarize the posterior distribution of ancestral ranges using the RRW model, we annotated nodes in a maximum clade credibility tree (MCC) using the program TreeAnnotator v1.7.5. This tree was then used as an input in SpreaD3 (*Bielejec et al., 2016*) to reconstruct the pattern of spatial diffusion and to visualize lineage diversification across the landscape.

**Table 1** Detection of geographic barriers for *Hyalinobatrachium fleischmanni* by multiple approaches based on genetic structure. (1) Monte Carlo test of Nei's pairwise *Fst* based on 999 replicates, (2) agreement between the border maps generated with Geneland for the considered barriers, (3) degree of overlap between the two first sPCA components between adjacent populations, and (4) supported monophyletic lineages between distinct geographical regions.

| Geographic barrier | Adjacent regions | $F_{ST}$ $(n = 115)$ | Geneland $(n = 60)$ | sPCA $(n = 115)$ | Phylogeny $(n = 115)$ |
|---|---|---|---|---|---|
| Eastern Cordillera | Serranía del Perijá/Magdalena | 0.0543 | no agreement | complete overlap | not monophyletic |
| Andes Range | Magdalena/Chocó | **0.3624**[**] | no agreement | complete overlap | not monophyletic |
| Central Panama Isthmus | Chocó/Central Panama | **0.3452**[**] | no agreement | complete overlap | not monophyletic |
| Western Panama Isthmus | Central Panama/ North Talamanca Range | **0.7509**[***] | **agreement** | **no overlap** | not monophyletic |
| Talamanca Range | North Talamanca Range/ South Talamanca Range | **0.9071**[**] | **agreement** | **no overlap** | **monophyletic** |
| Hess Escarpment | South Talamanca Range/Chortis block | 0.4055 | no agreement | partial overlap | not monophyletic |
| MPJ fault system | Chortis block/Maya block | **0.5262**[**] | **agreement** | **no overlap** | **monophyletic** |
| Isthmus of Tehuantepec | Maya block/southern Mexico block | **0.3733**[**] | **agreement** | **low overlap** | not monophyletic |

**Notes.**
Bold values show the support of tested barriers for each analysis.
[**]*p*-value $\leq$ 0.01.
[***]*p*-value $\leq$ 0.001.

If *H. fleischmanni* had a South American origin and subsequently dispersed to Central America after the closure of the Isthmus of Panama as we hypothesized, we would expect that the MRCA of all *H. fleischmanni* would be located in South America, and the more recent nodes would represent dispersal events into North America.

# RESULTS

We generated a final alignment of 2,036 base pairs (bp) for 123 samples from 9 countries, including 13 sequences obtained from the GenBank and BOLD system databases (Table S1). The obtained sequences, representing unique haplotypes of the single genes, are available via GenBank (accession numbers: MG944443–MG944695; Supplementary Material 2). We did not detect any stop codons in protein-coding genes (COI, ND1). We obtained 25 haplotypes with an overall haplotypic diversity $(h) = 0.863$ and a nucleotide diversity $(\pi) = 1.282$ for the 16S gene. In contrast, we found 63 haplotypes for COI, with $h = 0.979$ and $\pi = 0.044$, and 45 haplotypes for ND1, with $h = 0.991$ and $\pi = 0.042$ (Table S1.2).

## Spatial clustering analysis

The Nei's pairwise $F_{ST}$ results showed significant differences between adjacent regions for all barriers except for the HE and the Eastern Cordillera (Table 1). The Geneland map of population membership for the concatenated genes revealed nine clusters that partially coincided with our hypothesized barriers, depicting the IT, the MPJ fault system, the Talamanca Range and the WPI as barriers. Additionally, three non-considered barriers were indicated between samples from west of the IT and on the Chortis block, between samples from both sides of the Darien and between the Ecuadorian and Colombian samples (Fig. 2A).

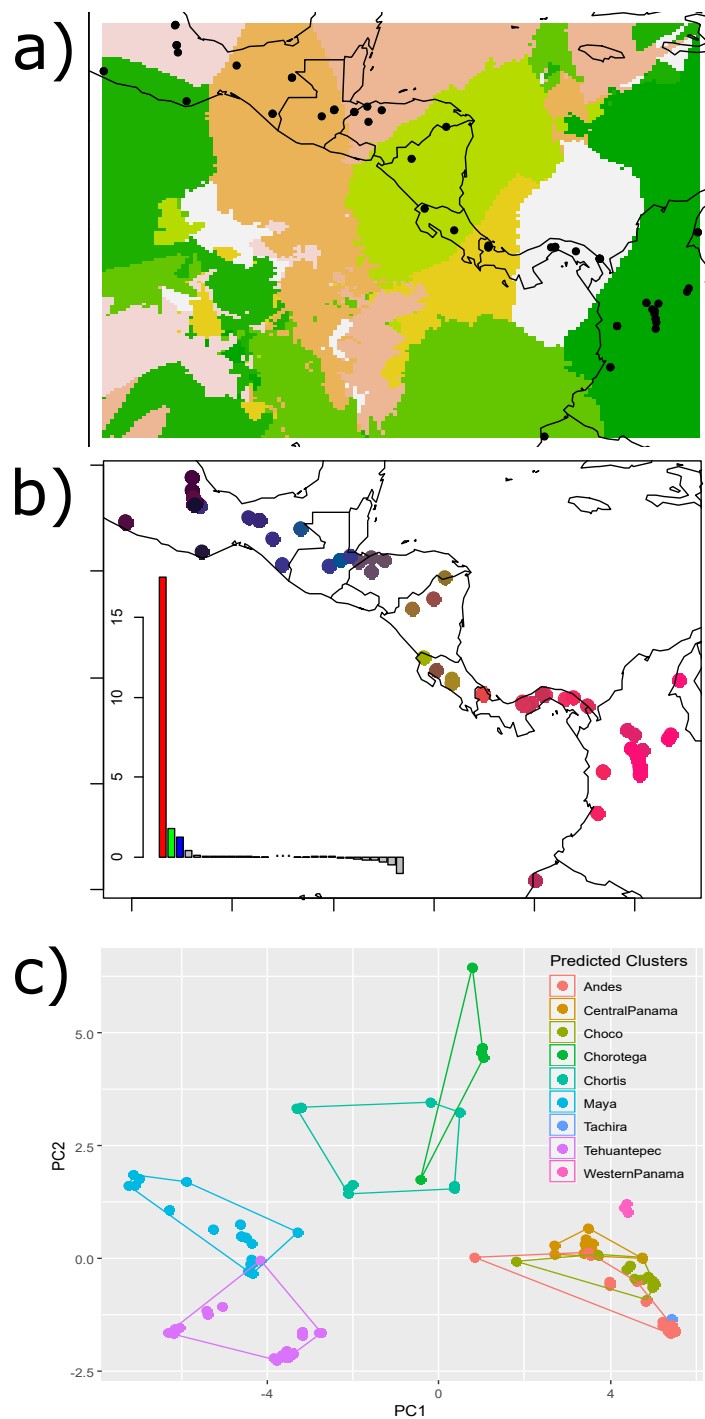

**Figure 2** **Results of the (A) Bayesian (Geneland) and (B–C) multivariate spatial analyses (sPCA) for *H. fleischmanni* population clustering based on mtDNA sequences.** Results of the (A) Bayesian (Geneland) and (B–C) multivariate spatial analyses (sPCA) for *H. fleischmanni* population clustering based on mtDNA sequences. For the sPCA analysis, the color of each point in (B) is determined in the red-green-blue (RGB) system based on the score of each individual on the first (translated to a red channel) and second axes (translated to green) of the sPCA. The points of (C) represent the values of the two first sPCA components, and the colors indicate the predicted clusters resulting from the hypothesized barriers.

sPCA performed on individual genotypes revealed a significant, geographically correlated genetic structure for all three genes (nper = 999, $P = 0.001$). Eigenvalues indicated a higher spatial genetic structure for the main axis, related to the global structure. The positive axis of the first sPCA (regional scale) exhibited the greatest variation in genetic distance in relation to the distance network (Fig. 2B). The samples separated by the WPI, the Talamanca Range and the MPJ fault system did not overlap based on the two sPCA first components, while the samples separating the IT and HE exhibited partial overlap (Fig. 2C).

## Phylogenetic patterns, times of divergence, and demography

The PartitionFinder output indicated that HKY+I+G, GTR+G, and GTR +G+I were the best models for 16S, COI, and ND1 respectively. The phylogenetic relationships based on the Bayesian and maximum likelihood approaches for the concatenated genes indicated three main, well-supported clades, although their relative positions were not fully resolved (Fig. 3). The first clade, designated the "Northern clade", was divided into two lineages: a large lineage containing all samples from the SMB and Maya regions (pp = 1) and a smaller one from the western Chortis region (pp = 1). The second clade, the "Central clade", was comprised of samples from the eastern Chortis and Chorotega regions (pp = 1). The third clade, the "Southern clade", consisted of samples from the Chocó and SAP regions, including the species *H. tatayoi* from Venezuela (pp = 1). The Southern clade did not show any structure, displaying a polytomic topology.

The divergence time estimation results showed a pattern of divergence among the three main clades occurring during the Pliocene (~3.40 Ma, HPD = 2.25–4.56 Ma; Fig. 3). With respect to the Northern clade, the split between the lineage from West Chortis and the remaining samples also occurred during the Pleistocene in the Gelasian age (~2.19 Ma, HPD = 1.38–3.53), while separation between samples from the Pacific and Gulf+Maya regions occurred at the beginning of the Calabrian age (~1.76 Ma, HPD = 0.86–2.23). Finally, the split between lineages from the Gulf and Maya regions occurred near the end of the Calabrian age (~1.51 Ma, HPD = 0.62–1.73). The divergence between the Central and Southern clades occurred at the end of the Pliocene (~2.64 Ma, HPD = 1.50–3.68), while splits within each clade began at the end of the Calabrian age for the Central clade (~0.81 Ma, HPD = 0.12–0.71) and during the Gelasian age (~1.64 Ma, HPD = 0.63–1.89) for the Southern clade (Fig. 3).

The 95% CIs of the effective population size (BSP results) overlapped along the entire time period in the Northern clade (Fig. 4A). However, the ancient and recent effective population sizes of the Southern clade did not overlap (by 95% CI of the Bayesian posterior probability), providing significant support for a change in population size and suggesting that the clade exhibited a constant population size and posterior expansion at approximately 0.1–0.3 Ma (Fig. 4B).

The results regarding Bayesian spatiotemporal diffusion (Fig. 5) highlight the Chorotega and West Chortis region as the most likely ancestral geographic area for *H. fleischmanni*, suggesting two subsequent dispersal events, in which the MRCA of the Northern clade was distributed in the environs of the Chortis and Maya regions, whereas that of the Southern clade (stem node) was distributed around the Chorotega and Chocó regions.

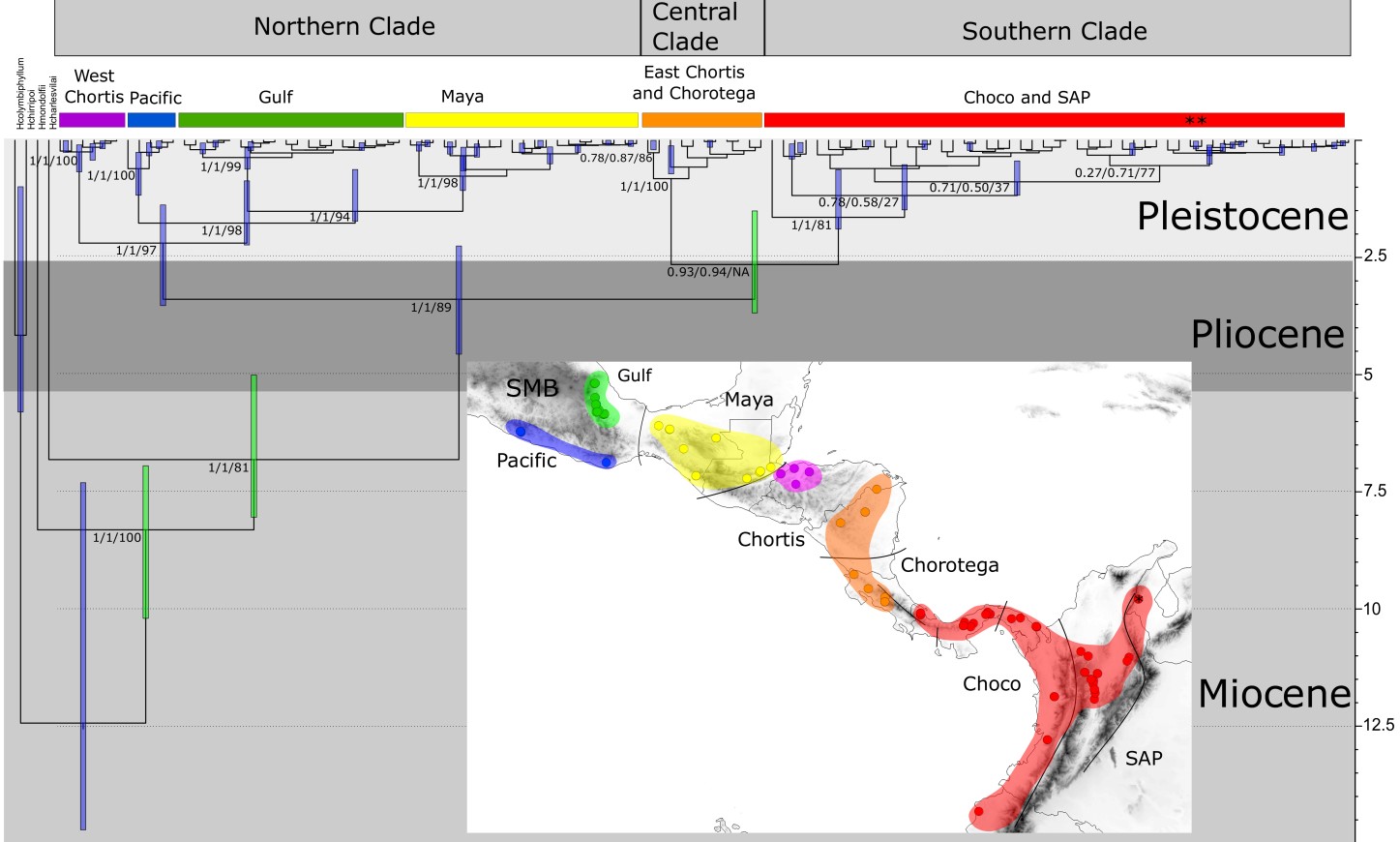

**Figure 3** **Time-calibrated tree of *H. fleischmanni* unique haplotypes.** Time-calibrated tree of *H. fleischmanni* unique haplotypes, inferred from BEAST based on the combined ribosomal (16S) and protein-coding (COI, ND1) mitochondrial sequences, with calibration on three nodes indicated by green bars (see 'Materials and Methods' for details). Blue rectangles over key nodes indicate the 95% highest posterior densities (HPD) of the estimated times of divergence events (in Ma). Clade support is indicated by *posterior* BI values in BEAST and MrBayes and by RAxML Bootstrap analysis and is presented in this order separated by a slash. Asterisks at tips represent *H. tatayoi* samples included in the analysis. The inner map shows the geographic locations of haplotype lineages. Each color in the map coincides with the haplogroup obtained in the phylogenetic reconstruction.

Our results also reflect independent dispersal for samples west of the MPJ fault system and later divergence of the three remaining clusters in the Northern clade around the IT. For the Central clade, an initial range in the Chorotega region south of the Talamanca range was observed, with posterior dispersal towards eastern Chortis. Furthermore, an ancestral range was detected in the Chocó block within the Southern clade with subsequent dispersal towards the south and east, reaching the lowlands east of the Andes range to the south, while two lineages dispersed northwards independently, reaching the southern limit of the Chocó region.

## Genetic diversity and structure

Our results provide evidence of four of the hypothesized barriers (WPI, Talamanca Range, MPJ fault system and IT) and identified two additional barriers: one east of the IT and one within the Chortis block. Hence, we identified seven genetically homogeneous

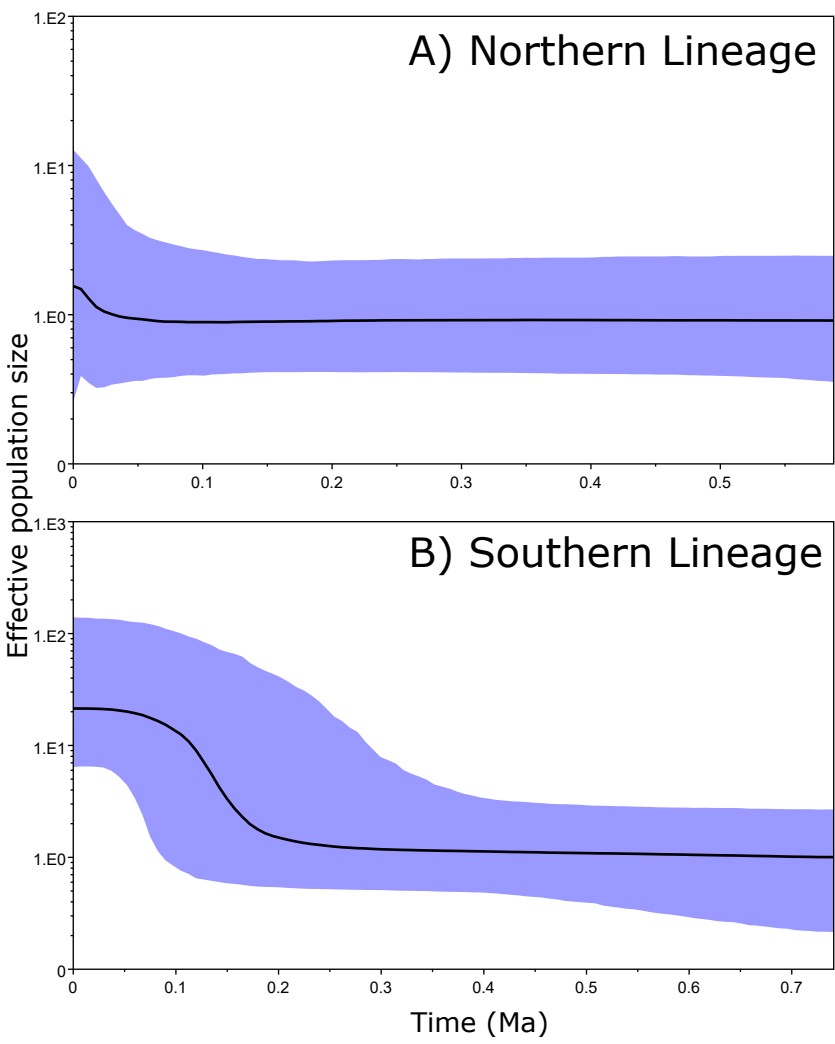

**Figure 4 Bayesian skyline plots.** Bayesian skyline plots for Northern (A) and Southern (B) clades generated through phylogenetic reconstruction.

regions: North American Pacific, Gulf of Mexico, Maya block, western Chortis, eastern Chortis-South Talamanca Range, North Talamanca Range and Chocó-SAP. A minimum genetic distance (Table 2) was exhibited between regions for North Talamanca Range and Chocó-SAP (K2P = 1.6%), while the maximum was observed between eastern Chortis and North Talamanca Range (K2P = 6.0%). Significant Nei's $F_{ST}$ indices between regions were obtained for almost all combinations ($F_{ST}$ >0.4), except between North Talamanca Range and Chocó-SAP (Table 2).

## DISCUSSION

The complex geologic and geographic history of Central America has long intrigued researchers, who have aimed to decipher how different features that act as barriers to or corridors for dispersal have affected the distribution and diversity of multiple taxa

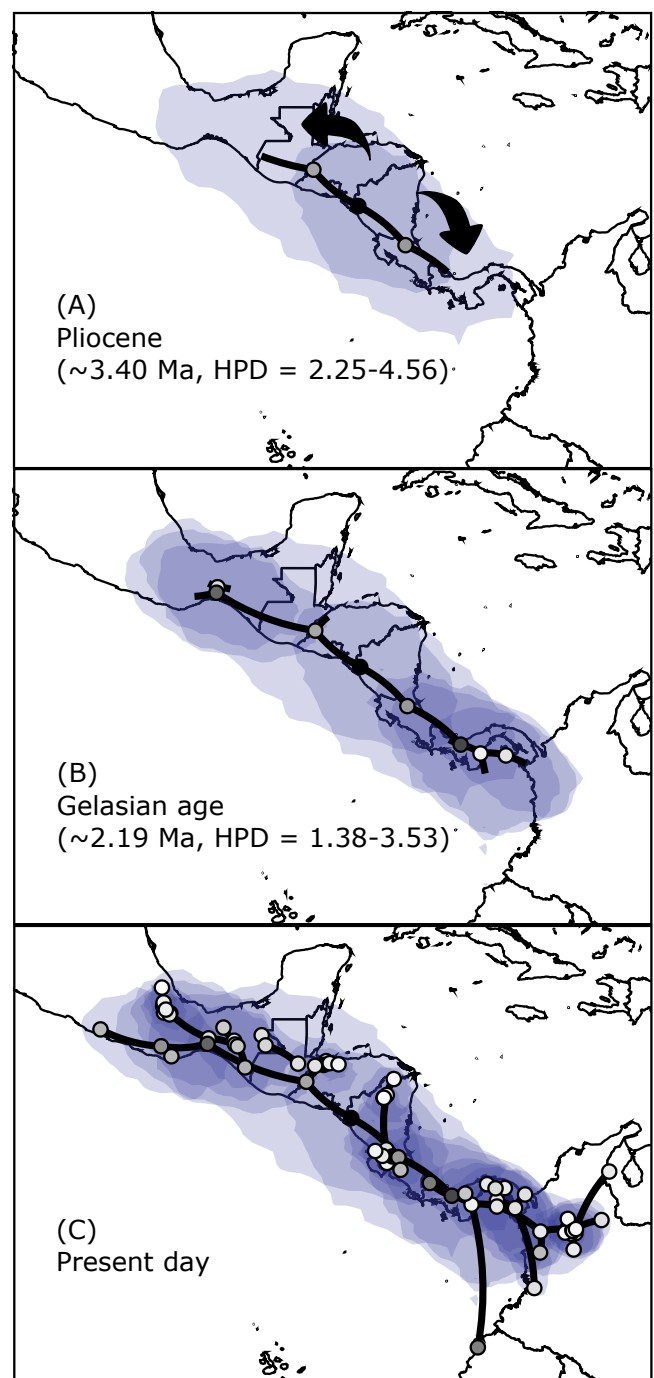

**Figure 5 Spatial projection of the Bayesian spatiotemporal diffusion analysis of *H. fleischmanni*.** Spatial projection of the Bayesian spatiotemporal diffusion analysis of *H. fleischmanni* lineages for three timepoints, based on the maximum clade credibility (MCC) tree estimated with a ''Relaxed Random Walk'' model. Lines represent branches of the MCC tree; shaded areas indicate the 95%-HPD uncertainty for the ancestral branches; the shading gradient indicates older (lighter) versus younger (darker) events; and dot color represents the ages of older nodes (darker) and younger tips (lighter).

**Table 2  Mean K2P distances within and among populations (lower) for all tree mitochondrial genes and Nei's pairwise $F_{ST}$ indices per pair of populations (upper).** Significance of $F_{ST}$ indices was estimated by a Monte Carlo test based on 999 permutations.

| | K2P Mean Within distance | Chorotega | Chortis | Gulf | Maya | Pacific | South America | Western Panama |
|---|---|---|---|---|---|---|---|---|
| Chorotega | 0.005 | – | 0.922[***] | 0.845[***] | 0.766[***] | 0.879[***] | 0.452[**] | 0.886[***] |
| Chortis | 0.004 | 0.048 | – | 0.801[***] | 0.687[***] | 0.876[***] | 0.432[**] | 0.950[***] |
| Gulf | 0.005 | 0.046 | 0.04 | – | 0.545[**] | 0.635[***] | 0.659[***] | 0.815[**] |
| Maya | 0.009 | 0.049 | 0.045 | 0.023 | – | 0.471[**] | 0.648[**] | 0.694[***] |
| Pacific | 0.004 | 0.047 | 0.036 | 0.022 | 0.023 | – | 0.413[*] | 0.848[***] |
| South America | 0.009 | 0.035 | 0.057 | 0.046 | 0.051 | 0.043 | – | 0.138 [ns] |
| Western Panama | 0.004 | 0.037 | 0.060 | 0.049 | 0.054 | 0.046 | 0.016 | – |

**Notes.**

ns, non-significant; *, $p$-value <0.05; **, $p$-value ≤ 0.01; ***, $p$-value ≤ 0.001.

(*Gutiérrez-García & Vázquez-Domínguez, 2013*; *Bagley & Johnson, 2014*). Our results show a deep phylogenetic structure of *H. fleischmanni*, which has differentiated as three well-supported clades, revealing old divergence events dating back to the Pliocene and younger divergence events within clades during the Pleistocene (Fig. 3). Additionally, our results agree with those of previous studies showing *H. fleischmanni* to be paraphyletic with *H. tatayoi* (*Guayasamin et al., 2008*; *Castroviejo-Fisher et al., 2011*; *Delia, Bravo-Valencia & Warkentin, 2017*), and we found no differences between the *H. tatayoi* and the *H. fleischmanni* samples within the South American clade.

## The Southern clade

The Southern clade encompasses samples from the Chocó and SAP regions, including the species *H. tatayoi* from Venezuela. Samples ranging from Panama to Venezuela and Ecuador grouped together, with no clear phylogenetic separation among them. However, haplotype networks and spatial clustering analyses for coding genes allowed us to identify a partition on both sides of the Northern Andes (Supplemental Information 1 and Fig. S3). The lack of significant structure for the Southern clade is remarkable, considering that the geographic distances between populations reach 1,600 km. In addition, the CPI and the Northern Andes, which are widely recognized as speciation drivers for both highland and lowland species (*Bagley & Johnson, 2014*; *Mendoza et al., 2015*), do not seem to have exerted any effect on the genetic structure of this clade. Notably, the genetic distance observed on both sides of the Northern Andes (COI K2P = 1.9%; Table S1.1) is lower than the distances reported for lowland species with a higher dispersal capacity from similar localities, such as the hummingbird *Amazilia amabilis* (COI K2P = 2.06%, *Mendoza et al., 2016*). These results contrast with previous knowledge of the ecology of glassfrog species, which have been found to be characterized by site fidelity (*Valencia-Aguilar, Castro-Herrera & Ramírez-Pinilla, 2012*), low mobility and restricted gene flow even at local scales (*Delia, Bravo-Valencia & McDiarmid, 2017*; *Robertson, Lips & Heist, 2008*). However, most of the previous research on this frog has focused on calling males and reproductive behavior (*Delia, Bravo-Valencia & McDiarmid, 2017*), while the dispersal capability of females and tadpoles, which can have a significant impact on mtDNA genetic structure, is still unknown.

Thus, it is possible that the Chocó and SAP regions present adequate conditions for tadpole dispersal, allowing range expansion. However, this hypothesis needs to be evaluated based on additional demographic studies and a greater sample size per site.

Our results do not support our initial hypothesis that the species was originally from South America and then dispersed through the Isthmus of Panama. Indeed, the Southern clade is rather young, and the various populations it encompasses differentiated during the last million years (middle Pleistocene). Our results show that this clade has experienced a recent population expansion during the last 100,000–300,000 years, reaching a relatively final stable population size, exhibiting a dispersal route from Central Panama to South America (Fig. 5). Based on the genetic and phylogenetic results for this clade, we suggest that its dispersal towards South America and on both sides of the Northern Andes occurred very recently, likely as a consequence of climatic oscillations during glacial periods (*Smith, Amei & Klicka, 2012*). Under this scenario, there has been insufficient time for effective genetic differentiation to occur, and the phylogenetic reconstruction therefore failed to resolve the divergence detected in the spatial analysis.

In particular, the lack of differentiation between sequences from east of Northern Andes (in the Magdalena Valley) and *H. tatayoi* in Serranía del Perijá is remarkable: sequences were identical with extremely low divergence ($F_{ST} = 0.0543$, Table 1), supporting a recent dispersal through the Eastern Cordillera. In contrast, we detected a slight structure for the Ecuadorian record (Geneland; Fig. 2B) based on a single specimen (QCAZ-22303). We acknowledge that a more detailed genetic structure analysis of the Colombian and Ecuadorian populations is needed, including samples from the southern departments in Colombia, and the use of additional genetic markers (e.g., microsatellites or SNPs) is suggested.

## The Northern clade

Unlike the Southern clade, the Northern clade shows significant genetic structure in four different lineages, ranging from western Chortis (Central America) to lowland forests in Veracruz and Guerrero (Mexico). One remarkable finding was the split between samples from either side of the MPJ fault system, where individuals separated by only 60 km exhibit great genetic distances (K2P = 4.5% for all three genes, Table 1), even reaching the limit of the 6% barcode gap for Neotropical amphibians in COI (5.2% for COI, Table S1.2; *Lyra, Haddad & Azeredo-Espin, 2017*). The calibration results showed that samples from these localities have been isolated since the Gelasian age (early Pleistocene; ~2.19 Ma, HPD = 1.38–3.53). The MPJ fault system has been recognized as the main barrier to dispersal for multiple species ranging from the Maya to the Chortis blocks (*Ornelas, Ruiz-Sánchez & Sosa, 2010*; *Barrera-Guzmán et al., 2012*; *Rovito et al., 2015*). Our results indeed support the hypothesis that MPJ has effectively acted as a barrier for *H. fleischmanni* dispersal.

On the other hand, the Geneland and phylogenetic results showed a more complex scenario, in agreement with the presence of three well-defined lineages within the Northern clade, one of which is located in the north of the SMB, another southward the SMB, and the last in the Maya block (Fig. 2B). This genetic structure is very similar to that observed for

the brush-finch *Arremon brunneinucha*, distributed in humid montane forests (*Navarro-Sigüenza et al., 2008*). Samples from the Gulf of Mexico and Pacific are clearly isolated by the highlands of the Sierra Madre Oriental and the Sierra Madre Occidental (Fig. 1), indicating a divergence pattern that is frequently detected for lowland species (*Mulcahy, Morrill & Mendelson, 2006*; *Rivera-Ortíz et al., 2016*) and species groups (*Streicher et al., 2014*; *Palacios et al., 2016*).

Within this clade, we found that samples from SMB do not group as a single lineage but instead display a paraphyletic position in relation to the lineage from the Maya region. These two regions present the lowest K2P among all comparisons performed in this analysis, except for samples separated by the WPI. Importantly, unlike the known impact of the WPI and CPI on lowland species, the effect of the IT has mostly been defined in relation to montane species (*Bryson, García-Vázquez & Riddle, 2011*; *Jiménez & Ornelas, 2016*), for which it represents a dispersal-limiting barrier. Currently, the IT is occupied by dry, scrubby coastal plains that are very different from the moist areas on either side (*Rodríguez-Gómez, Gutiérrez-Rodríguez & Ornelas, 2013*), so one plausible explanation for this finding is that the dry forests along the IT did not always act as a barrier to *H. fleischmanni*. Instead, considering that those clades were likely isolated between 1.76 to 2.19 Ma (HPD = 0.86–3.53), it is likely that the successive glacial cycles during the Pleistocene could have enabled the montane forests to reach lower elevations through the impact of climate change (*Barber & Klicka, 2010*), thus creating a temporal corridor and allowing the species to disperse through the region. Hence, for species like *H. fleischmanni*, the IT probably acted as both a corridor and a barrier, which is supported by our Bayesian spatiotemporal diffusion results (Fig. 5).

## The Central clade

Our results revealed a Central clade without any deep geographic structure expanding across the HE and separated from the Southern clade around the end of the Pliocene (~2.64 Ma). We did not find any evidence suggesting differentiation between samples from the Chortis and Chorotega regions, leading us to reject the hypothesis that the HE acted as a barrier. However, we must consider the small sample sizes from Honduras and Nicaragua ($n = 10$ samples), which likely limits detailed structural evaluation for this region. Most phylogeographic studies performed in the region known as nuclear Central America face similar problems, with limited or null sampling from northern Honduras (*Castoe et al., 2003*; *Crawford & Smith, 2005*; *Mulcahy, Morrill & Mendelson, 2006*) or sampling that is biased towards the dry forests of the Pacific coast (*Parkinson, Zamudio & Greene, 2000*; *Hasbún et al., 2005*; *Vázquez-Miranda, Navarro-Sigüenza & Omland, 2009*; *Poelchau & Hamrick, 2011*), where *H. fleischmanni* has not been recorded. Our Bayesian spatiotemporal diffusion results showed rapid dispersal from the Chorotega to Chortis blocks, with no apparent impact on the genetic structure of these populations (Fig. 5). Nevertheless, additional work is needed to confirm whether the main geographic features present in this region have driven the dispersal of low-mobility species such as *H. fleischmanni* in humid forests.

## Phylogeographic patterns

The three main clades that we identified for *H. fleischmanni* show deep intraspecific divergence, with genetic distance values (K2P) greater than 3% (Table 2). Indeed, the landscape analysis, Bayesian spatiotemporal diffusion analyses, and estimated divergence times revealed interesting patterns that allowed us to reconstruct the historical biogeography of these frogs and identify the impact of different geographic barriers on the genetic structure and phylogeographic patterns of *H. fleischmanni*. Although the main phylogenetic topology and the three major clades were well supported in Bayesian analyses, the maximum likelihood phylogenetic reconstruction did not resolve these relationships (see Supplemental Information 1).

The Bayesian spatiotemporal diffusion analysis did not support the hypothesis proposed by *Castroviejo-Fisher et al. (2014)* of a South American origin and subsequent dispersal to Central America. In contrast, *H. fleischmanni* appears to have originated in the region encompassing the Chorotega and eastern Chortis elements. Interestingly, we found that *H. fleischmanni* has undergone two dispersal events: one southwards towards the Chocó region and one northwards, reaching the Maya region, followed by vicariance events driven by the effect of the Chortis highlands and the Talamanca Range. Considering that all species that are closely related to *H. fleischmanni* are endemic to South America and that the divergence times among the three clades are similar (i.e., the isolation of the Northern clade occurred ca. 3.40 Ma (HPD = 2.25–4.56 Ma), while that between the two other clades occurred 2.64 Ma (HPD = 1.50–3.68)), it is likely that the MRCA ancestor of *H. fleischmanni* and *H. carlesvilai* was located in South America around 7.65 Ma (CI = 5.93–9.63). Later, the ancestor of *H. fleischmanni* arrived in Central America during the Pliocene, soon after the closure of the Isthmus of Panama (*Montes et al., 2015*), whereas the current South American populations are descendants of a second migration from Central to South America. Accordingly, the dispersal-vicariance events among the main clades potentially occurred simultaneously or over a very short time, which might explain why the position of the clades and their internal structure were not consistent between the Bayesian and maximum likelihood approaches.

Regarding the vicariance events for the Central and Southern clades, multiple elements need to be revised. The samples from each cluster that are geographically closest are located on opposite sides of the Talamanca Cordillera in the Chorotega block. The time of divergence of the MRCA for these clades (3.28 Ma, HPD = 1.59–3.86) coincides with the estimated age of the intervening mountains (1–2 Ma; *Denyer, Alvarado & Aguilar, 2000*; *Marshall et al., 2003*, which are recognized as a main driver of speciation (*Savage, 2002*). The time of divergence also coincides with the rise of the sea level during the mid-late Pliocene (~3.5–3 Ma), which generated a seaway, likely reinforcing the WPI break and therefore acting as a barrier across the Pacific region (*Cronin & Dowsett, 1996*; *Bagley, Hickerson & Johnson, 2018*). Hence, the central mountain ranges on Costa Rica and Panama and the eustatic sea levels around the WPI might have increased divergence, as documented for multiple spatial divergence patterns of amphibian species (*Crawford, Bermingham & Polanía-S, 2007*; *Wang, Crawford & Bermingham, 2008*; *Bagley, Hickerson & Johnson, 2018*).

The isolation of the Northern clade from the other two does not entirely correspond to our hypothesis of geographical barriers. Populations from both the Northern and Central clades are distributed throughout the Chortis block, indicating that the MPJ fault system was not the main driver of divergence between clades. On the other hand, our structure (Geneland) results suggest a frontier at the center of the Chortis block, near northeastern Honduras (Fig. 2B). Similar divergence patterns have been observed between two water-dependent subspecies of *Caiman crocodilus* (*Venegas-Anaya et al., 2008*), in agreement with the eastern limit of the Chortis highlands (*Morrone, 2014*; *Townsend, 2014*). Here, the complex topography resulting from multiple volcanic activities along the Chortis highlands during the last 2 Ma and the presence of dry habitats in the Pacific region (*Savage, 2002*) may have isolated the *H. fleischmanni* populations during the late Pliocene. This hypothesis is in agreement with the high species endemism recognized for the region (*Anderson et al., 2010*; *Townsend et al., 2012*), in which intensive study is required to evaluate the underestimated regional taxonomic diversity (*Townsend & Wilson, 2016*).

## Taxonomic implications

Previous studies have suggested that *H. fleischmanni* is a paraphyletic species in relation to its sister species *H. tatayoi* (*Castroviejo-Fisher et al., 2011*; *Delia, Bravo-Valencia & Warkentin, 2017*). Here, we confirm the paraphyly of the species, as the *H. tatayoi* samples are grouped within the Southern clade, lacking significant genetic differences from the western Andes samples. Furthermore, our overall Bayesian topology agrees with the results obtained by *Delia, Bravo-Valencia & Warkentin (2017)* for 12S sequences. We identified three main isolated lineages with large genetic distances that likely include cryptic diversity within Central America. To confirm the existence of cryptic—and potential candidate—species, different lines of evidence must be obtained; in fact, we are conducting a follow-up study of the integrative taxonomy of this species complex that includes genetic, morphologic and acoustic data (Mendoza et al., 2018, unpublished data) to describe lineage divergence and determine the identities and geographic distributions of all valid species (*Padial et al., 2010*).

## CONCLUSIONS

We have conducted the most comprehensive analysis of genetic variation and divergence within *H. fleischmanni* to date, producing one of the few phylogenetic and phylogeographic studies for glassfrogs, with the exception of a few studies from Guyana (*Castroviejo-Fisher et al., 2011*; *Jowers et al., 2015*), and this is the first such study of a Central American species. Moreover, our results aided in the successful reconstruction of the historical biogeography of these frogs and dispersal and vicariance events during the history *H. fleischmanni* lineages, revealing a higher complexity for the species than expected, especially for the Northern lineage, in which significant population structure was found. Indeed, our results support the Talamanca range, the MPJ fault system, and the Chortis highlands as significant factors exerting effects on the dispersal of lowland amphibians during the late Pliocene and early Pleistocene. Additionally, we suggest that the IT acted as both a corridor and a barrier for *H. fleischmanni* during the early Pleistocene, while the HE and the Andes Range

did not act as significant barriers. The complementary use of phylogenetic and landscape analyses allowed us to perform an adequate evaluation of dispersal patterns and potential barriers within this region; hence, our approach can be applied in biogeographic and phylogeographic studies of different taxa.

## ACKNOWLEDGEMENTS

We sincerely thank the following researchers and institutions for their kind contributions of tissue samples: David Wake from the Museum of Vertebrate Zoology (Berkeley, USA), Juan M. Daza from Museo de la Universidad de Antioquia (Colombia), Fernando Rojas-Runjaic from Museo de Historia Natural La Salle (Venezuela), Beatriz Alvarez from Museo Nacional de Ciencias Naturales (CSIC, España), María Estefanía López and Andres M. Cuervo from Colección de Tejidos of Instituto Alexander von Humboldt (Colombia), Adrian Nieto from Museo de Zoología de la Facultad de Ciencias (UNAM, México), Jhon Tailor Rengifo from Universidad Tecnológica del Chocó (Colombia), Carlos R. Vásquez-Almazán from Universidad de San Carlos de Guatemala (Guatemala), Colección de Docencia (Universidad del Valle, Colombia), and Josiah Townsend from Indiana University of Pennsylvania. We also thank Aldo Lopez, Mirna G. Garcia, Angel F. Soto Omar Becerra, Itzue Caviendes-Solis, Angel I. Contreras, Nelson Martín Cerón de la Luz, Abigail Mora-Reyes, Rene Murrieta-Galindo, Hibraim Pérez-Mendoza, Alejandro Rodriguez, Jose Criollo, Eliana Barona, Oscar Cuellar and Jonard David Echavarria-Renteria for their keen assistance in collecting samples and Mailyn A. Gonzalez, Eduardo Tovar Luque, Andrea Jiménez, Martin García and Laura Cifuentes for allowing the use of laboratory facilities and their invaluable support in laboratory procedures. An earlier version of the manuscript was improved due to the kind suggestions of Victoria Sosa, Juan Manuel Guayasamin, Cristiano Vernesi and one anonymous reviewer. This paper constitutes a partial fulfillment of the Graduate Program in Biological Sciences (Posgrado en Ciencias Biológicas) of the Universidad Nacional Autónoma de México (UNAM) for AMM.

### Funding

This project was supported by PAPIIT-DGAPA (UNAM) project IN203617 and the Rufford Foundation (Rufford Small Grant reference 18423-1). The funders had no role in study design, data collection and analysis, decision to publish, or preparation of the manuscript.

### Grant Disclosures

The following grant information was disclosed by the authors:
PAPIIT-DGAPA (UNAM) project: IN203617.
Rufford Foundation (Rufford Small Grant): 18423-1.

### Competing Interests

Gabriela Parra Olea is an Academic Editor for PeerJ.

## Author Contributions

- Angela M. Mendoza conceived and designed the experiments, performed the experiments, analyzed the data, contributed reagents/materials/analysis tools, prepared figures and/or tables, authored or reviewed drafts of the paper, approved the final draft.
- Wilmar Bolívar-García and Roberto Ibáñez contributed reagents/materials/analysis tools, authored or reviewed drafts of the paper, approved the final draft.
- Ella Vázquez-Domínguez and Gabriela Parra Olea conceived and designed the experiments, contributed reagents/materials/analysis tools, authored or reviewed drafts of the paper, approved the final draft.

## Field Study Permissions

The following information was supplied relating to field study approvals (i.e., approving body and any reference numbers):

Ministerio de Medio Ambiente, Colombia gave permission for specimen collection (Resolution 120 of 24 August 2015) and Secretaría del Medio Ambiente y Recursos Naturales in Mexico for Collection Permit 00947/16.

## Data Availability

Sequences are provided in Table S1.

## Supplemental Information

Supplemental information for this article can be found online at http://dx.doi.org/10.7717/peerj.6115#supplemental-information.

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
