# Peer review of "The role of Central American barriers in shaping the evolutionary history of the northernmost glassfrog, Hyalinobatrachium fleischmanni (Anura: Centrolenidae)"

_PeerJ, doi:10.7717/peerj.6115_

## Round 0.1 · original submission · Major Revisions

I hope that you can deal with two crucial issues that need to be taken into account. The first deals with sampling: only a small number of individuals from most localities or pooled “populations” were collected and as one of the reviewers indicated even collections from Venezuela and Ecuador are scarce. The second deals with analyses, the three markers are mitocondrial thus they are linked, and some of the analyses performed need to be carried out with more markers. If you are able to modify objectives and analyses considering what you have (one molecular marker and a small sampling) please modify as well hypotheses.

·

Basic reporting

- Supplementary figures should be improved as follows: (i) Species names should be added. (ii) Terminals should not be overlapping.

- Table S1 needs improvement as follows: Add “species”, “elevation”, “Genbank code” and “source” as columns in this table. The table should include outgroups. “Source” should include the studies that generated the sequences.

Experimental design

- Most of the results and discussion are concentrated on a gene-by-gene analysis. However, all used genes are mitochondrial and, therefore, are linked. It is my opinion that the methods should focus on the joined analyzes of the three genes together.

- There are very few sequences from Venezuela and Ecuador. This might have an impact on the low genetic structure found by the authors.

- An additional potential dispersal barrier in South America is the Tachira Depression.

Validity of the findings

- ¿Could the low genetic structure found in South America be associated to the reduced number of terminals included, specially from Venezuela and Ecuador?

- Authors should mention the possibility of having cryptic diversity in Central America.

- The authors should discuss in more detail the taxonomic status of H. tatayoi.

- The authors should discuss a little bit more about the geographic origin of H. fleischmanni. Note that all species that are closely related to H. fleischmanni are endemic to South America. Thus, a South American origin of fleischmanni and dispersal to Central America makes a lot of biogeographic sense. The authors need to be explicit about this biogeographic patterns, and make their case for a Central American origin.

Additional comments

The ms is a very good contribution for our understanding of phylogeographic pattern and, potentially, diversification forces in both Central and South America. As mentioned before, however, I think that the study needs some reanalysis and a deeper discussion on some aspects.

I have included all my comments on a Word file (attached).

Reviewer 2 ·

Basic reporting

The manuscript is, for the most part, well-written and presents a mitochondrial dataset with very nice geographic coverage for a wide-ranging frog species. My two main concerns are described below in ‘Experimental design’.

Minor revisions throughout the manuscript are recommended to simplify overwritten sentences and would help improve clarity for readers. Some examples:

Line 21 – “has been useful for developing an understanding of the effects of different processes and barriers on the” could be simply “has greatly affected the distribution” or alternatively “has been useful for understanding processes that influence the distribution…”

Line 28 – suggested change: “three mitochondrial regions, two coding (COI and ND1) and one ribosomal (16S)”

Lines 164-165 – suggested change: “We performed spatial clustering analyses to identify patterns consistent with dispersal barriers.”

As written currently, the results are not clearly connected to the hypotheses. Lines 121-128 state several hypotheses, but it is not clear how these are tested. What results would indicate a South American origin and dispersal into Central America? What results would provide evidence that mountains are acting as barriers? What are the expectations for genetic structure if sea level change has had an impact? These predictions and outcomes should be integrated into the appropriate methods and results sections.

The authors have done a great and thorough job with data reporting, providing museum codes for vouchers, locality info, and GenBank accession numbers.

Experimental design

The two main areas for improvement relate to 1) a lack of explicit hypothesis testing, and 2) the reliance on effectively a single locus, mitochondrial DNA. The second concern is addressed in lines 152-161 indicating the authors are aware of potential shortcomings, but this section needs some improvement.

Specifically, the line “we could only perform our analyses…” should be rephrased to something like “we chose to perform our analyses with gene sequences that could be directly compared”. However, I am also unclear what exactly the authors mean by direct comparison. Do they refer to the GenBank/BOLD sequences they included, and thus did not have tissue for? Or are they referring to comparisons made about divergence timing or something else that might benefit from using common mitochondrial genes?

The main area of improvement would be to better connect the hypotheses with the analyses that were conducted. The last methods section suggests that hypothesis testing was done in some way, but it is very unclear. How were barriers “evaluated” or “tested”? Were these a priori regions tested using some sort of phylogenetic constraint analysis? I think this would greatly strengthen the paper if so.


Additional suggestions for improvement:

Line 180-182 – In my opinion, it would be more straightforward to combine the data (and just include missing data for some individual/locus combinations). I don’t think the differences in Fig. 2 are meaningful, since the mtDNA is effectively a single locus. The small differences are more likely related to the number of informative sites in each gene.

Line 183 – Clarify that by “landscape analyses” you mean the spatial clustering analyses conducted above, if true. Since there are some differences between methods and genes, how were the groups assigned?

The haplotype networks are somewhat redundant with clustering and phylogenetic inferences, all done with the same mitochondrial genes. Could be moved to the supplement.

Line 309-311 – Clarification needed - 95% credible intervals overlapped or did not overlap with what?

Validity of the findings

The conclusions are reasonable, but as described above, should be better presented to integrate the objectives, hypotheses, results, and interpretation.

---

## Round 0.2 · Minor Revisions

Please pay special attention to the suggestion of Dr. Vernesi to present an haplotype network as it is usually shown in phlylogeographic research. The size of the circles is related to the frecuency of the haplotypes. Also consider some additional explanations that this reviewer ask specifically.

·

Basic reporting

No comment.

Experimental design

No comment.

Validity of the findings

No comment.

Additional comments

I think that the authors have made a really good effort to improve the manuscript. This study shows very nicely the effect of biogeographic barriers on genetic structure and provides valuable information that may be related to diversification processes.

·

Basic reporting

I noticed that the authors provided GenBank number accessions in Table S1. According to me, it would be better to report in the main text too that sequences generated in this study have been already submitted to GenBank.

Overall, the paper reads well and it is clear, fulfilling the four required criteria.

Experimental design

It is evident that the paper has been significantly improved after the first revision. The authors took into proper and serious considerations criticism raised by the reviewers.

However, in my opinion there are still at least two major issues that need to be fixed:
1. L326 and Table 2. Please, report the associated p-values to Fst. There's no point in discussing a differentiation index if it is unknown whether it is statistically significant.

2. I might be wrong, that no 'true' haplotype network is reported in the current version of the manuscript. In fact, what presented in Figures S1 and S2 are again phylogenetic trees. Haplotype network are meant to show the frequency of different haplotypes (with circle size proportional to frequency) and relationships among them in terms of number of mutations separating each haplotype in a kind of, loosely speaking, parsimony context.
I think that a phylogeographic investigation would greatly benefit from showing a proper haplotype network. User friendly software such as PopArt (http://popart.otago.ac.nz/index.shtml) are available for inferring haplotype network from sequence alignment.

Validity of the findings

No comment

Additional comments

The following minor points need to be considered:

1. L107. Can you, please, name some of the related species?

2. Please, provide adequate explanation for **

---

## Round 0.3 · accepted · Accept

Thank you for considering suggestions by the two reviewers. The only issue I am not sure is whether in Figure legends you should spell the name of the genus. However I am sure that the editorial staff will indicate whether or not it is necessary to include the full name of the genus.

#